# Synthesis, Structural Characterization and Ligand-Enhanced Photo-Induced Color-Changing Behavior of Two Hydrogen-Bonded Ho(III)-Squarate Supramolecular Compounds

**DOI:** 10.3390/polym11081369

**Published:** 2019-08-19

**Authors:** Chih-Chieh Wang, Szu-Yu Ke, Yun Feng, Mei-Lin Ho, Chung-Kai Chang, Yu-Chun Chuang, Gene-Hsiang Lee

**Affiliations:** 1Department of Chemistry, Soochow University, Taipei 11102, Taiwan; 2Instrumentation Center, National Taiwan University, Taipei 10617, Taiwan; 3Materials Science Group, National Synchrotron Radiation Research Center, Hsinchu 30076, Taiwan

**Keywords:** coordination polymer, metal-organic framework, hydrogen bond, supramolecular architecture, color change

## Abstract

Two coordination polymers (CPs) with chemical formulas, [Ho_2_(C_4_O_4_)_2_(C_2_O_4_)(H_2_O)_8_]·4H_2_O (**1**) and [Ho(C_4_O_4_)_1.5_(H_2_O)_3_] (**2**), (C_4_O_4_^2−^ = dianion of squaric acid, C_2_O_4_^2−^ = oxalate), have been synthesized and their structures were determined by single-crystal X-ray diffractometer (XRD). In compound **1**, the coordination environment of Ho(III) ion is eight-coordinate bonded to eight oxygen atoms from two squarate, one oxalate ligands and four water molecules. The squarates and oxalates both act as bridging ligands with μ_1,2_-*bis*-monodentate and *bis*-chelating coordination modes, respectively, connecting the Ho(III) ions to form a one-dimensional (1D) ladder-like framework. Adjacent ladders are interlinked via O–H⋅⋅⋅O hydrogen bonding interaction to form a hydrogen-bonded two-dimensional (2D) layered framework and then arranged orderly in an AAA manner to construct its three-dimensional (3D) supramolecular architecture. In compound **2**, the coordination geometry of Ho(III) is square-antiprismatic eight coordinate bonded to eight oxygen atoms from five squarate ligands and three water molecules. The squarates act as bridging ligands with two coordination modes, μ_1,2,3_-*tris*monodentate and μ_1,2_-*bis*-monodentate, connecting the Ho(III) ions to form a 2D bi-layered framework. Adjacent 2D frameworks are then parallel stacked in an AAA manner to construct its 3D supramolecular architecture. Hydrogen bonding interactions between the squarate ligands and coordinated water molecules in **1** and **2** both play important roles on the construction of their 3D supramolecular assembly. Compounds **1** and **2** both show remarkable ligand-enhanced photo-induced color-changing behavior, with their pink crystals immediately turning to yellow crystals under UV light illumination.

## 1. Introduction

Lanthanide metal-organic frameworks (LnMOFs) have received much attention, not only for their fascinating structural variety, but also for their potential applications of MOFs like porosity for gas storage [1,2,3,4,5,6,7,8,9], their specific characteristics arising from 4f electrons for luminescence [10,11,12,13,14,15,16,17,18,19,20,21,22]. The inherent character of lanthanide ions with high affinity for oxygen atoms and high coordination numbers, result in the formation of a number of MOFs with flexible coordination geometry and various structural dimensionality from multi-carboxylate ligands [13,14,15,16,17,18,19,20,21,22,23,24,25,26,27,28,29]. The squarate, C_4_O_4_^2−^, has been widely used as a polyfunctional ligand, including (1) acts as a bridging ligand with various coordination modes (μ_2_ to μ_6_ bridges shown in Scheme 1) to build up many coordination polymers with novel extended networks, including 1D chain, 2D layer, 3D cube- and cage-like frameworks and so forth and (2) behaviors as hydrogen bond donor, acceptor or π−π constructor for the assembly of extended supramolecular architecture [30,31,32,33,34,35,36,37,38,39,40,41,42,43,44,45,46,47,48,49,50,51,52,53,54,55,56,57,58,59,60,61,62,63]. In the previous investigation, several 2D and 3D LnMOFs constructed via the bridges of lanthanide and squarate ligand with various coordination modes have been synthesized under hydrothermal or solvothermal conditions [64,65,66,67,68,69,70]. Their thermal behavior, magnetic property and photo-luminescence spectra of 2D and 3D LnMOFs have also been studied. With our continuous effort on the study of metal-squarate coordination polymers (CPs) [58,59,60,61,62,63], we report here the synthesis, structural characterization, thermal stability and light-induced color-changing behavior of two Ho(III)-squarate hydrogen-bonded supramolecular networks, [Ho_2_(C_2_O_4_)(C_4_O_4_)_2_(H_2_O)_8_]·4H_2_O (**1**) and [Ho(C_4_O_4_)_1.5_(H_2_O)_3_] (**2**), (C_4_O_4_^2−^ = dianion of squaric acid, C_2_O_4_^2−^ = dianion of oxalic acid), in which the squarate acts as the bridging ligands with μ_1,2_-coordination mode (Scheme 1b) for **1** and combined μ_1,2_-plus μ_1,2,3_-coordination modes (Scheme 1b,c) for 2 to build up 1D ladder-like CP and 2D bi-layered MOF, respectively.

## 2. Materials and Methods

### 2.1. Materials and General Methods

All the reagents (Sigma-Aldrich Inc., Taipei, Taiwan) were purchased commercially and used without further purification. Elementary microanalyses (EA) (C, H and N) were performed using a Perkin-Elmer 2400 elemental analyzer (PerkinElmer, Taipei, Taiwan). FTIR spectra (500–4000 cm^−1^) were recorded from KBr pellets with a Nicolet Fourier Transform IR, MAGNA-IR 500 spectrometer (ThermoFisherScientific, Waltham, MA, USA). Thermal analysis (TGA) was carried out using a Perkin-Elmer 7 Series/UNIX TGA7 analyzer (PerkinElmer, Taipei, Taiwan) under a nitrogen atmosphere in the temperature range of 25 °C–700 °C with a ramp rate of 5 °C/min. 

### 2.2. Synthesis of [Ho_2_(C_2_O_4_)(C_4_O_4_)_2_(H_2_O)_8_]·4H_2_O (**1**) and [Ho(C_4_O_4_)_1.5_(H_2_O)_3_]·H_2_O (**2**)

Method 1: Squaric acid (H_2_C_4_O_4_, 8.6 mg, 0.075 mmol) was dissolved in 3 mL mixed solvents of distilled water and EtOH (1:1, *v/v*) and then added into the solution of Ho(NO_3_)_3_⋅5H_2_O (22.1 mg, 0.05 mmol) and 4,4’-bipyridyl-N,N’-dioxide hydrate (bpno, 9.4 mg, 0.05mmol) in 6 mL mixed solvents of distilled water and EtOH at room temperature to give a colorless solution. Light-pink needle-like and block-like crystals of **1** and **2**, respectively, were obtained after several weeks in 2.69% and 47.3% yields. The resulting crystals were collected by filtration, washed several times with distilled water and dried in air. 

Method 2: The synthetic procedure was similar to method 1 except Na_2_C_2_O_4_ was added into the reaction solution, with the molar ratios of squaric acid (H_2_C_4_O_4_, 8.5 mg, 0.075 mmol), Ho(NO_3_)_3_⋅5H_2_O (44.1 mg, 0.1 mmol), 4,4’-bipyridyl-N,N’-dioxide hydrate (bpno, 9.4 mg, 0.05mmol) and Na_2_C_2_O_4_ (10.1 mg, 0.075 mmol) in 12 mL mixed solvents of distilled water and EtOH at room temperature. Only light-pink needle-like crystals of **1** were obtained after four weeks in 66.4% yields. The resulting crystals were collected by filtration, washed several times with distilled water and dried in air.

Anal. Calcd for HoC_5_H_12_O_12_ (**1**) (Mw=429.07): C 14.00, H 2.82. Found: C 13.92, H 2.81. IR (KBr pellet): ν = 3340 (s), 1681 (s), 1607 (s), 1536 (s), 1477 (vs), 1317 (m), 1098 (m), 869 (m), 830 (m), 659 (m), 537 (m), 494 (m) cm^−1^. Anal. Calcd for HoC_6_H_8_O_10_ (**2**) (Mw = 405.05): C 17.79, H 1.99 Found: C 17.68, H 1.85. IR (KBr pellet): 3420 (s), 3101 (m), 1605 (s), 1509 (vs), 1097 (m), 858 (m), 741 (m), 671 (m), 645 (m) cm^−1^.

### 2.3. Crystallographic Data Collection and Refinements

Single crystals of **1** and **2** suitable for X-ray structural analyses were selected and their crystallographic data were collected at 150 K and 250 K, respectively, on a Siemens SMART CCD diffractometer using Mo radiation (*λ* = 0.71073 Å) in the ω scan mode. Cell parameters were retrieved using SMART [71] software and the detector frames were integrated by use of program SAINT [72]. Data reduction was performed by use of program SAINT [73] and corrected for Lorentz and polarization effects. The empirical absorption corrections were performed using the SADABS [73] program. Both the structures were solved by direct method and refined by full-matrix, least-squares procedures using the SHELXTL-PC V 5.03 software [74]. All non-hydrogen atoms were refined subjected to anisotropic refinements. The hydrogen atoms of the coordinated and solvated water molecules were located in the Difference Fourier map with the corresponding positions and isotropic displacement parameters being refined. The final full-matrix, least-squares refinement on *F*^2^ was applied for all observed reflections (I > 2σ(I)). Details of crystallographic data, data collections and structure refinements **1** and **2** are summarized in Table 1. CCDC 1940248 and 1940249 for **1** and **2**, respectively.

### 2.4. In Situ Powder X-ray Diffraction

The powder X-ray diffraction patterns of **1** and **2** were measured at the BL01C2 in National Synchrotron Radiation Research Center (NSRRC). The wavelength of the incident X-rays was 1.0332 Å (12.0 keV) and diffraction signals were recorded with a Mar345 imaging-plate detector. The powder sample was packed into a 0.3 mm diameter glass capillary. The diffraction geometry was calibrated by NIST standard reference material, lanthanum hexaboride (SRM660b). The one-dimensional diffraction pattern was converted with GSAS-II program [75].

### 2.5. Spectral Measurement

UV-vis diffusive reflectance spectra for compounds **1** and **2** were obtained with a HITACHI U-3900H spectrophotometer (Hitachi High Technologies America Inc., Schaumburg, IL, USA) equipped with an integrating sphere accessory (Al_2_O_3_ was used as a reference) [76].

## 3. Results and Discussion

### 3.1. Syntheses and Characterization of Compounds **1** and **2**

Compounds **1**, [Ho(C_2_O_4_)_0.5_(C_4_O_4_)(H_2_O)_4_]·2(H_2_O) and **2**, [Ho(C_4_O_4_)_1.5_(H_2_O)_3_], were synthesized by direct mixing of Ho(NO_3_)_3_⋅5H_2_O, squaric acid (H_2_C_4_O_4_) and 4,4’-bipyridyl-N,N’-dioxide (bpno) in the mixed solvents of distilled water and EtOH at room temperature (method 1) as shown in Scheme 2. In addition, compound **1** can also be obtained by the direct mixing of Ho(NO_3_)_3_⋅5H_2_O, H_2_C_4_O_4_, bpno and Na_2_C_2_O_4_ in the mixed solvents of distilled water and EtOH (method 2). The yields of **1** are increasing from 2.69% to 66% but the crystals quality is not as good as those obtained by method 1. The bpno may act as a base for the deprotonation of the squaric acid. In the absence of bpno, compounds **1** and **2** were not produced during the reaction. In method 1, the oxalate ligand was obtained via the in-situ synthesis from the squarate ligand, indicating a slow release of oxalate from the squarate could be helpful for the formation of compound **1**. However, the formation mechanism of oxalate is not clear and may be through an in situ oxidation ring-opening reaction of squarate. This type of ring opening oxidation reaction has been proposed previously [77,78,79]. The most relevant IR features are those related to the bridging oxalate and squarate ligands. Strong bands are found centered at around 1681, 1607, 1536, 1477 and 1605, 1509 cm^−1^ for **1** and **2**, respectively, which are attributed to the vibration modes of the C=O and mixtures of C–O and C−C stretching motions. They are in agreement with the characteristic of the oxalate and (CO)*_n_*^2−^ salts [80]. Additional broad peaks for **1** and **2** appear in the region of 3100–3500 cm^−1^, corresponding to the stretching vibration of ν(O−H) from water molecules.

### 3.2. Structure Description of {[Ho(C_2_O_4_)_0.5_(C_4_O_4_)(H_2_O)_4_]·2H_2_O}_n_ (**1**)

The molecular structure of **1**, shown in Figure 1a, reveals that the Ho(III) ion is eight coordinate bonded with two oxygen donors from two squarate (C_4_O_4_^2−^) ligands, two oxygen donors from one oxalate (C_2_O_4_^2−^) ligand and four water molecules, with Ho(III)–O distances in the range of 2.314(6)–2.470(6). The related bond distances and angles around the Ho(III) ion are listed in Table 2. The C_4_O_4_^2−^ acts as a bridging ligand with μ_1,2_-*bis*-monodentate coordination mode (Scheme 1b) connecting the Ho(III) ions to form a one-dimensional (1D) linear chain. Two linear chains are interlinked via the bridges of the Ho(III) ions and C_2_O_4_^2−^ ligand with *bis*-chelating coordination mode, forming a 1D ladder-like CP (Figure 1b). Intra-ladder O−H⋅⋅⋅O type hydrogen bonds between the coordinated water molecules and uncoordinated oxygen of C_4_O_4_^2−^ provide extra energy on the stabilization of these 1D CPs. The Ho⋅⋅⋅Ho separations bridged by the C_4_O_4_^2−^ and C_2_O_4_^2−^ ligands are 7.696(1) and 6.372(5) Å, respectively. It is worth to note that hydrogen bonding interaction in **1** play an important role on the extension of 1D ladder-like chains to 2D layered networks and then a 3D supramolecular architecture, as shown in Figure 1c,d. Firstly, adjacent 1D ladder-like polymeric chains are mutually interlinked via two inter-ladder O−H⋅⋅⋅O type hydrogen bonds between the coordinated water molecules (O(8) and O(9)) and uncoordinated oxygen atoms (O(3) and O(4)) of C_4_O_4_^2−^ ligands, with O⋅⋅⋅O distance of 2.761(10) and 2.675(10) Å, respectively, extended to a two-dimensional (2D) hydrogen-bonded layered network (Figure 1c). Adjacent 2D hydrogen-bonded layers are arrayed in an orderly AAA stacking manner and then extended to its 3D supramolecular network (Figure 1d,e) via the intermolecular O−H⋅⋅⋅O type hydrogen bonds among the coordinated water molecules and the oxygen atoms of carbonyl groups of C_4_O_4_^2−^ and C_2_O_4_^2−^ ligands (yellow dashed lines shown in Figure 1d), with the O⋅⋅⋅O distance in the range of 2.677(10) ~ 2.825(10) Å. The 1D hydrophilic pores surrounded by squarate and oxalate ligands are generated with the pore size of 6.37 × 7.70 Å, as viewing along *a* axis (Figure 1e). Two guest water molecules (O(11) and O(12)) are intercalated into the 1D hydrophilic pores in the 3D supramolecular network (Figure 1e) and further stabilized via four O−H⋅⋅⋅O type hydrogen bonds between the oxygen atoms of C_4_O_4_^2−^ and C_2_O_4_^2−^ ligands in 1D polymeric chains and the guest water molecules. The related parameters of hydrogen bonds in **1** are listed in Table 3.

### 3.3. Structure Description of [Ho(C_4_O_4_)_1.5_(H_2_O)_3_]_n_ (**2**)

The molecular structure of **2**, shown in Figure 2a, reveals that the Ho(III) ion is eight coordinate in a square antiprismatic geometry bonded with three oxygen donors from three μ_1,2,3_-squarates, two oxygen donors from two μ_1,2_-squarates and three water molecules with Ho(III)–O distances in the range of 2.305(5)–2.415(5). The related bond distances and angles around the Ho(III) ions are listed in Table 4. The C_4_O_4_^2−^ acts as a bridging ligand with two types of coordination modes, μ_1,2__,3_-*tris*-monodentate (Scheme 1c) and μ_1,2_-*bis*-monodentate (Scheme 1b) coordination modes, in which the first one connect the Ho(III) centers to form a two-dimensional (2D) layered framework (Figure 2b, left). Two layers are mutually interlinked via the bridges of the Ho(III) ions and disorder μ_1,2_-*bis*-monodentate C_4_O_4_^2−^ (Figure 2b, right), forming a 2D bi-layered MOF (Figure 2b, center). The Ho⋅⋅⋅Ho separations bridged by the μ_1,2__,3_-*tris*-monodentate C_4_O_4_^2−^ ligand are 6.399(1), 6.486(1) and 8.132(1) Å and bridged by the μ_1,2_-*bis*-monodentate C_4_O_4_^2−^ ligand is 6.508(4) Å, respectively. Similar to **1**, hydrogen bonding interaction in **2** also play an important role on the extension of 2D bi-layered MOFs to a 3D supramolecular architecture, as shown in Figure 2c. Adjacent 2D bi-layered MOFs are arrayed in an orderly AAA stacking manner and extended to its 3D supramolecular network (Figure 2c,d) via the intermolecular O−H⋅⋅⋅O type hydrogen bonds among the coordinated water molecules and the oxygen atoms of C_4_O_4_^2−^ (yellow dashed lines in Figure 2c,d) with the O⋅⋅⋅O distance in the range of 2.668(5)–2.914(5) Å. The related parameters of hydrogen bonds are listed in Table 5.

Compared **1**, **2** and the other Ho(III)-squarate polymeric framework, [Ho_2_(C_4_O_4_)_3_(H_2_O)_4_]*_n_* (**3**), synthesized under solvothermal condition reported in the previous literature [66]. The Ho(III) ions in **1** an **2** are both eight-coordinate, but, in **3**, is nine coordinate with a tricapped trigonal prismatic coordination environment. The inherent character of Ho(III) ion with high affinity for oxygen atoms and high coordination numbers [13,14,15,16,17,18,19,20,21,22,23,24,25,26,27,28,29], result in the formation of Ho(III)-squarate coordination polymers with flexible coordination geometry and various structural dimensionality. The squarate act as bridging ligands with μ_1,2_-bis-monodentate (Scheme 1b) coordination mode in **1**, μ_1,2__,3_-tris-monodentate (Scheme 1c) and μ_1,2_-bis-monodentate (Scheme 1b) coordination modes in **2** and bidentate/monodentate μ_3_− (Scheme 1g) and bidentate/monodentate μ_4_– (Scheme 1i) coordination modes in **3**, connecting the Ho(III) ions forming 1D chain, 2D bi-layer and 2D network structures, respectively. The numbers of oxygen atoms of squarate ligand bonded to the Ho(III) ion in **1**, **2** and **3** are 2, 5 and 5, respectively. The oxalate ligands in **1**, instead of squarate ligands, bonded to the Ho(III) ion in a bis-chelating bridging mode connect two Ho(III)-squarate chain forming a 1D ladder-like polymeric framework, which generate 1D hydrophilic pores for the accumulation of guest water molecules in the 3D supramolecular architecture. It is important to note that both the coordinated and guest water molecules play important roles on the construction of their 3D supramolecular architectures and further stabilized via the intermolecular O−H⋅⋅⋅O hydrogen bonds among the squarate or oxalate ligands, coordinated and guest water molecules.

### 3.4. Thermal Stability of CPs **1** and **2**

In order to investigate the thermal stability and structural variation of compounds **1** and **2**, thermogravimetric analysis (TGA) and in-situ temperature dependent XRD measurements were performed as shown in Figure 3 and Figure 4, respectively. During the heating process, the TGA of **1** (Figure 3a) revealed that a two-steps weight-losses were observed with the first weight loss of 24.7% occurred in the range of approximate 47–269 °C, corresponding to the losses of coordinated and solvated water molecules (calc. 25.2%) and then thermal stable up to 375 °C without any weight loss. On further heating, samples decomposed at approximately 375–700 °C. The TGA of **2** (Figure 4a) revealed that **2** is thermally stable up to 95 °C and then a two-step weight-losses were observed with the first weight loss of 17.4% occurred in the range of approximate 95–197 °C, corresponding to the loss of coordinated water molecules (calc. 17.0%) and then thermal stable up to 403 °C without any weight loss. On further heating, these samples decomposed at approximately 403–700 °C. To gain the structural changes as a function of the temperature, in situ powder XRD patterns of **1** and **2** were performed and the results at several specific temperatures were shown in Figure 3b and Figure 4b, respectively. Based on the result of TGA, the guest and coordinated water molecules in **1** are lost in the first-step weight loss. The subtle relative intensity varies between RT data and simulation pattern, it may be due to the composition change of solvated and coordinated water molecules. Above 140 °C, the crystallinity becomes worse and worse. As the temperature rising from 170 °C to 200 °C, the framework structure collapsed. The powder XRD patterns of **2** was shown in Figure 4b. The pattern at RT is almost identical to the simulation one obtained from single-crystal X-ray diffraction data. As the temperature rising to 200 °C, a phase transition occurred and can be sustained at 440 °C. As the temperature above 470 °C, **2** decomposed to an amorphous phase. All of the in-situ PXRD measurements are in agreement with the TGA.

### 3.5. UV-Visible Spectroscopy of CPs **1** and **2**

The solid-state adsorption spectra of [Ho(C_2_O_4_)_0.5_(C_4_O_4_)(H_2_O)_4_]·2(H_2_O) (**1**) and [Ho(C_4_O_4_)(H_2_O)_3_]*_n_* (**2**) were investigated at room temperature. As shown in Figure 5, the adsorption spectra bands of [Ho(C_2_O_4_)_0.5_(C_4_O_4_)(H_2_O)_4_]·2(H_2_O) (**1**, black line) and [Ho(C_4_O_4_)_1.5_(H_2_O)_3_]*_n_* (**2**, black line) both shows peaks at 361, 386, 418, 451, 468, 474, 486, 537 and 642 nm which can be ascribed to the (^3^H_6_, ^5^G_5_) ← ^5^I_8_, ^3^K_7_ ← ^5^I_8_, ^5^G_5_ ← ^5^I_8_, (^5^F_1_, ^5^G_6_) ← ^5^I_8_, ^3^K_8_ ← ^5^I_8_, ^5^F_2_ ← ^5^I_8_, ^5^F_3_ ← ^5^I_8_, (^5^F_4_, ^5^S_2_) ← ^5^I_8_ and ^5^F_5_ ← ^5^I_8_ transitions of the Ho^3+^ ion, respectively [81,82]. Interestingly, Figure 5a–d) also show reversible color changes immediately and UV-Vis spectra of **1** and **2** under illumination from an incandescent source/daylight to a LED light with a cellphone. The color change between pink (Figure 5a of **1** and Figure 5c of **2** and light yellow (Figure 5b of **1** and Figure 5d of **2** of the Ho^3+^ ion is caused by two absorption bands: (^5^F_1_, ^5^G_6_) ← ^5^I_8_ and (^5^F_4_, ^5^S_2_) ← ^5^I_8_. The ^5^G_6_ ← ^5^I_8_ transition at 447 nm is a so called “hypersensitive transition,” which intensity is dependent on the local surrounding of the holmium ion in symmetry and the ligand type [81]. Accordingly, in comparison to the Figure 5a,c, **1** and **2** have an enhanced adsorption in the region around 450–480 nm (red line in the UV spectra), indicating that the ligand may enhance the absorption of whole coordination polymers.

## 4. Conclusions

In conclusion, two 3D supramolecular frameworks, [Ho_2_(C_2_O_4_)(C_4_O_4_)_2_(H_2_O)_8_]·4(H_2_O) (**1**) and [Ho(C_4_O_4_)_1.5_(H_2_O)_3_] (**2**), have been successfully synthesized under a facile one-pot synthetic route and their structural versatility of the Ho(III) ion bridged by C_4_O_4_^2−^ ligands have been studied in detail. The high affinity for oxygen atoms and high coordination numbers of Ho(III) ions result in the formation of eight-coordinate environments bonded to oxygen atoms of two squarate, one oxalate and four water molecules in **1** and five squarate and three water molecules in **2**, respectively. In **1**, both the squarate and oxalate act as bridging ligands adopting μ_1,2_-*bis*-monodentate and *bis*-chelating coordination modes, respectively, connecting the Ho(III) ions forming the 1D ladder-like CPs, which generates hydrophilic pores intercalated guest water molecules. In **2**, the squarate acts as bridging ligand with two coordination modes, μ_1,2_-*bis*-monodentate and μ_1,2,3_-tris-monodentate, connecting the Ho(III) ions forming 2D bi-layered MOFs. Intermolecular hydrogen bonds among the squarate, oxalate ligands and coordinated, guest water molecules provide the main force on the structural extension from their 1D ladder-like CP or 2D layered MOF to 3D supramolecular architectures. The solid-state adsorption spectra of **1** and **2** both show reversible color-changing images under illumination from an incandescent source/daylight to a LED light with a cellphone. Both **1** and **2** have an enhanced adsorption in the region around 450-480 nm (red line in the UV spectra), indicating that the ligand may enhance the absorption of whole coordination polymers.

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
