# Peer review of "Synthesis, Structural Characterization and Ligand-Enhanced Photo-Induced Color-Changing Behavior of Two Hydrogen-Bonded Ho(III)-Squarate Supramolecular Compounds"

_polymers, 2019, doi:10.3390/polym11081369_

Round 1
Reviewer 1 Report
The manuscript by C-C. Wang et al. reports the synthesis, structural and thermal characterization of Holmium-squarate coordination polymer. In addition, the mixed-ligand squarate-oxalate coordination polymer was isolated and characterized. As follows from the experimental part, this mixed-ligand CP was obtained as an admixture to the main product 2, so it seems to me that the only product with reproducible synthesis is compound 2. Although the manuscript is clearly written, in my opinion, synthesis and characterization of one new coordination polymer are not enough to justify the publication in Polymers. If the authors will choose to revise their manuscript, they could consider the following remarks:
1. Extensive literature citation in the introduction part, 70 references for the first and only paragraph of the introduction part.
2. It is not clear how the crystals of products 1 and 2 were separated, probably manually, which is not a good laboratory practice.
3. Since the yield of compound 1 was only 2-3 % it is advisable to try to optimize the reaction conditions in order to obtain pure compound 2. Perhaps in the course of optimization, the authors would find conditions for selective formation of compound 1 also.
4. Line 223 indicates Figure 5 to be in the supporting information while actually, it is in the main part of the manuscript.
5 The color change should also be shown in UV-Vis spectra plots, which are absent in the manuscript.
6. One of possible holmium-squarate coordination polymer was reported before in reference 66 of the manuscript. I suggest giving a brief comparison of the structure reported in the manuscript with the previously known.
Author Response
Reviewer 1
The manuscript by C-C. Wang et al. reports the synthesis, structural and thermal characterization of Holmium-squarate coordination polymer. In addition, the mixed-ligand squarate-oxalate coordination polymer was isolated and characterized. As follows from the experimental part, this mixed-ligand CP was obtained as an admixture to the main product 2, so it seems to me that the only product with reproducible synthesis is compound 2. Although the manuscript is clearly written, in my opinion, synthesis and characterization of one new coordination polymer are not enough to justify the publication in Polymers. If the authors will choose to revise their manuscript, they could consider the following remarks:
Extensive literature citation in the introduction part, 70 references for the first and only paragraph of the introduction part.Answer: Thanks for the reviewer’s suggestion. In the introduction part, 70 references include the introduction of Ln-MOFs, the various coordination modes of squarate on the construction of M-squarate structures and related Ln-squarate MOFs. We think they are suitable and enough for the introduction part.
It is not clear how the crystals of products 1 and 2 were separated, probably manually, which is not a good laboratory practice.
Answer: Thanks for the reviewer’s comment. The crystal shapes of compounds 1 and 2 are different. 1 is needle-like crystals but 2 is block-like crystals. They can be separated easily by picking up the crystals under the microscopy. They can also be obtained by two synthetic methods, which are described in the revised manuscript.
Since the yield of compound 1 was only 2-3 % it is advisable to try to optimize the reaction conditions in order to obtain pure compound 2. Perhaps in the course of optimization, the authors would find conditions for selective formation of compound 1
Answer: Thanks for the reviewer’s comment. Actually, we have tried many methods to optimize the reaction condition, include changing the synthetic ratio and reaction time. But compounds 1 and 2 were co-crystallized in the solution by using method 1. Fortunately, the yields of 1 are low and the crystal shapes are different. So, they can be separated easily. In addition, 1 can be obtained by method 2, which is described in the experimental and result and discussion sections in the revised manuscript.
Line 223 indicates Figure 5 to be in the supporting information while actually, it is in the main part of the manuscript.
Answer: Sorry for the mistake. This error has been corrected in the revised manuscript.
The color change should also be shown in UV-Vis spectra plots, which are absent in the manuscript.
Answer: Thanks for the reviewer’s comment. The color changes of compounds 1 and 2 in UV-Vis spectra have been revised in Figure 5. Also the Figure has been attached as follows.
Figure 5. The color-changing images and UV spectra of 1 (a) & (b) and 2 (c) & (d).
One of possible holmium-squarate coordination polymer was reported before in reference 66 of the manuscript. I suggest giving a brief comparison of the structure reported in the manuscript with the previously known.
Answer: Thanks for the reviewer’s suggestion. The brief comparison of the structures among 1, 2, and the holmium-squarate coordination polymer in the ref. 66 is added in the revised manuscript and described as follow: “Compared 1, 2 and the other Ho(III)-squarate polymeric framework, [Ho2(C4O4)3(H2O)4]n (3), synthesized under solvothermal condition reported in the previous literature[66]. The Ho(III) ions in 1 an 2 are both eight-coordinate, but, in 3, is nine coordinate with a tricapped trigonal prismatic coordination environment. The inherent character of Ho(III) ion with high affinity for oxygen atoms and high coordination numbers[13-29], result in the formation of Ho(III)-squarate coordination polymers with flexible coordination geometry and various structural dimensionality. The squarate ligand act as bridging ligands with m1,2-bis-monodentate (Scheme 1(b)) coordination mode in 1, m1,2,3-tris-monodentate (Scheme 1(c)) and m1,2-bis-monodentate (Scheme 1(b)) coordination modes in 2 and bidentate/monodentate m3- (Scheme 1(g)) and bidentate/monodentate m4- (Scheme 1(i)) coordination modes in 3, connecting the Ho(III) ions forming 1D chain, 2D bi-layer and 2D network structures, respectively. The numbers of oxygen atoms of squarate ligand bonded to the Ho(III) ion in 1, 2 and 3 are 2, 5 and 5, respectively. The oxalate ligands in 1, instead of squarate ligands, bonded to the Ho(III) ion in a bis-chelating bridging mode connect two Ho(III)-squarate chain forming a 1D ladder-like polymeric framework, which generate 1D hydrophilic pores for the accumulation of guest water molecules in the 3D supramolecular architecture. It is important to note that both the coordinated and guest water molecules play important roles on the construction of their 3D supramolecular architectures and further stabilized via the intermolecular O-H×××O hydrogen bonds among the squarate or oxalate ligands, coordinated and guest water molecules.”

Reviewer 2 Report
The manuscript presented by Wang and co-workers show a clear and simple pathway to obtain Ho+3 metal-organic frameworks. The employed characterization techniques provide enough evidence of the nature of synthesized coordination compounds, but some issues must be solved before pass to a second review round.
Minor issues
1. In the introductory section it is very recommendable that authors include a scheme with the structure of the ligands and the possible coordination modes obtained with them.
2. In scheme 2, the addition of a draw of the monomer that generate each corresponding polymeric could clarify the differences observed on the supramolecular structure
3. On figures 1 and 2, it is necessary to include additional figures that allow appreciate the 3D arrangements, particularly those that show the channels or holes formed.
4. On figure 1 the authors mention that water molecules identified as O11 and O12 are intercalated in the micropores generated and participate in the stabilization of the polymeric structure through formation of hydrogen bonds, please include the figure showing that interactions.
Mayor issues
1. A very good description of the main intermolecular forces that stabilize the supramolecular structures formed by Compound 1 [Ho(C2O4)0.5(C4O4)(H2O)4]·2(H2O), and 2, [Ho(C4O4)1.5(H2O)3], was provided. However, scarce information about the micropores, channels or pockets formed by these compounds was include, this topic must be deeply described. The authors focus their discussion to the hydrogen bonds that promote the generation of 1D and 2D structures, but no discussion is provided about the 3D structures generated, the influence of Ho3+ coordination sphere modification by the inclusion of oxalate instead squarate ligand, the role played by the coordinated and non-coordinated water molecules on that 3D structure and their possible capacity to hold small molecules or metal ions in this cavities. The 3D structures obtained with these Ho3+ coordination compounds, provide different/similar cavities to that obtained by other Ho3+ derivatives with different coordination spheres? Please enhance the discussion about this topic.
2. In section 3.5 UV-visible spectroscopy of CPs, the spectra of both polymers must be included and the corresponding electronic transitions assigned. The photographs provided by the authors are very helpful, but the information obtained from the spectra could provide could be correlated with the different stability nicely described in section 3.4.Also, the authors state that “In comparison to the Ho(III) salts, 1 and 2 have a remarkable color changes, indicating that the ligand may enhance the absorption of whole coordination polymers”…but not indicate or suggest how this absorption enhance was produced. Are there different electronic transitions or the absorption intensity is modified on CPs compared with Ho3+ salts? If there are, identify them in the spectra.
3. The conclusions must be modified considering the above suggestions
Author Response
Reviewer 2
The manuscript presented by Wang and co-workers show a clear and simple pathway to obtain Ho+3 metal-organic frameworks. The employed characterization techniques provide enough evidence of the nature of synthesized coordination compounds, but some issues must be solved before pass to a second review round.
Minor issues
In the introductory section it is very recommendable that authors include a scheme with the structure of the ligands and the possible coordination modes obtained with them.Answer: Thanks for the reviewer’s suggestion. The structure of the squarate ligand and its possible coordination modes are shown in Scheme 1.
In scheme 2, the addition of a draw of the monomer that generate each corresponding polymeric could clarify the differences observed on the supramolecular structure.
Answer: Thanks for the reviewer’s suggestion. The monomers of compounds 1 and 2 are added in Scheme 2.
On figures 1 and 2, it is necessary to include additional figures that allow appreciate the 3D arrangements, particularly those that show the channels or holes formed.
Answer: Thanks for the reviewer’s suggestion. The 3D arrangements of 1 and 2 are added in revised manuscript and shown in Figure 1(e) and Figure 2(c) & 2(d).
On figure 1 the authors mention that water molecules identified as O11 and O12 are intercalated in the micropores generated and participate in the stabilization of the polymeric structure through formation of hydrogen bonds, please include the figure showing that interactions.
Answer: Thanks for the reviewer’s suggestion. The guest water molecules (O11 and O12) intercalated in the channels generated and participate in the stabilization of the polymeric structure through formation of hydrogen bonds are shown in Figure 1(e) and Table 3 in the revised manuscript.
Mayor issues
A very good description of the main intermolecular forces that stabilize the supramolecular structures formed by Compound 1, [Ho(C2O4)0.5(C4O4)(H2O)4]·2(H2O), and 2, [Ho(C4O4)1.5(H2O)3], was provided. However, scarce information about the micropores, channels or pockets formed by these compounds was include, this topic must be deeply described. The authors focus their discussion to the hydrogen bonds that promote the generation of 1D and 2D structures, but no discussion is provided about the 3D structures generated, the influence of Ho3+coordination sphere modification by the inclusion of oxalate instead squarate ligand, the role played by the coordinated and non-coordinated water molecules on that 3D structure and their possible capacity to hold small molecules or metal ions in this cavities. The 3D structures obtained with these Ho3+ coordination compounds, provide different/similar cavities to that obtained by other Ho3+ derivatives with different coordination spheres? Please enhance the discussion about this topic.Answer: Thanks for the reviewer’s suggestion. The brief description of the structures among 1, 2, and the holmium-squarate coordination polymer in the ref. 66 is added in the revised manuscript and described as follow: “Compared 1, 2 and the other Ho(III)-squarate polymeric framework, [Ho2(C4O4)3(H2O)4]n (3), synthesized under solvothermal condition reported in the previous literature[66]. The Ho(III) ions in 1 an 2 are both eight-coordinate, but, in 3, is nine coordinate with a tricapped trigonal prismatic coordination environment. The inherent character of Ho(III) ion with high affinity for oxygen atoms and high coordination numbers[13-29], result in the formation of Ho(III)-squarate coordination polymers with flexible coordination geometry and various structural dimensionality. The squarate ligand act as bridging ligands with m1,2-bis-monodentate (Scheme 1(b)) coordination mode in 1, m1,2,3-tris-monodentate (Scheme 1(c)) and m1,2-bis-monodentate (Scheme 1(b)) coordination modes in 2 and bidentate/monodentate m3- (Scheme 1(g)) and bidentate/monodentate m4- (Scheme 1(i)) coordination modes in 3, connecting the Ho(III) ions forming 1D chain, 2D bi-layer and 2D network structures, respectively. The numbers of oxygen atoms of squarate ligand bonded to the Ho(III) ion in 1, 2 and 3 are 2, 5 and 5, respectively. The oxalate ligands in 1, instead of squarate ligands, bonded to the Ho(III) ion in a bis-chelating bridging mode connect two Ho(III)-squarate chain forming a 1D ladder-like polymeric framework, which generate 1D hydrophilic pores for the accumulation of guest water molecules in the 3D supramolecular architecture. It is important to note that both the coordinated and guest water molecules play important roles on the construction of their 3D supramolecular architectures and further stabilized via the intermolecular O-H×××O hydrogen bonds among the squarate or oxalate ligands, coordinated and guest water molecules.”
In section 3.5 UV-visible spectroscopy of CPs, the spectra of both polymers must be included and the corresponding electronic transitions assigned. The photographs provided by the authors are very helpful, but the information obtained from the spectra could provide could be correlated with the different stability nicely described in section 3.4.Also, the authors state that “In comparison to the Ho(III) salts, 1 and 2 have a remarkable color changes, indicating that the ligand may enhance the absorption of whole coordination polymers”…but not indicate or suggest how this absorption enhance was produced. Are there different electronic transitions or the absorption intensity is modified on CPs compared with Ho3+ salts? If there are, identify them in the spectra.
Answer: We appreciate the reviewer’s comment and suggestion. The color changes of compounds (1) and (2) in UV-Vis spectra have been revised in Figure 5. Also the Figure has been attached as follows. The reason of the color change has also been added to the section of 3.5 and below.
“The solid-state adsorption spectra of [Ho(C2O4)0.5(C4O4)(H2O)4]·2(H2O) (1) and [Ho(C4O4)(H2O)3]n (2) were investigated at room temperature. As shown in Fig. 5, the adsorption spectra bands of [Ho(C2O4)0.5(C4O4)(H2O)4]·2(H2O) (1, black line) and [Ho(C4O4)1.5(H2O)3]n (2), black line) both shows peaks at 361, 386, 418, 451, 468, 474, 486, 537 and 642 nm which can be ascribed to the (3H6, 5G5) ← 5I8, 3K7 ← 5I8, 5G5 ← 5I8, (5F1, 5G6) ← 5I8, 3K8 ← 5I8, 5F2 ← 5I8, 5F3 ← 5I8, (5F4, 5S2) ← 5I8 and 5F5 ← 5I8 transitions of the Ho3+ ion, respectively.[81-82] Interestingly, Fig. 5(a)-(b) and Fig. 5(c)-(d) also show reversible color changes immediately and UV-Vis spectra of 1 and 2 under illumination from an incandescent source/daylight to a LED light with a cellphone. The color change between pink (Fig. 5(a) of 1 and 5(c) of 2 and light yellow (Fig. 5(b) of 1 and 5(d) of 2 of the Ho3+ ion is caused by two absorption bands: (5F1, 5G6) ← 5I8 and (5F4, 5S2) ← 5I8. The 5G6 ← 5I8 transition at 447 nm is a so called “hypersensitive transition”, which intensity is dependent on the local surrounding of the homium ion in symmetry and the ligand type.[81] Accordingly, in comparison to the 5(a) and 5(c), 1 and 2 have an enhanced adsorption in the region around 450-480 nm (red line in the UV spectra), indicating that the ligand may enhance the absorption of whole coordination polymers.”
Figure 5. The color-changing images and UV spectra of 1 (a) & (b) and 2 (c) & (d).
The conclusions must be modified considering the above suggestions.
Answer: Thanks for the reviewer’s suggestion. The conclusion gas been modified as follows: “In conclusion, two 3D supramolecular frameworks, [Ho(C2O4)0.5(C4O4)(H2O)4]·2(H2O) (1) and [Ho(C4O4)1.5(H2O)3] (2), have been successfully synthesized under a facile one-pot synthetic route and their structural versatility of the Ho(III) ion bridged by C4O42- ligands have been studied in details. The high affinity for oxygen atoms and high coordination numbers of Ho(III) ions result in the formation of eight-coordinate environments bonded to oxygen atoms of two squarate, one oxalate and four water molecules in 1 and five squarate and three water molecules in 2, respectively. In 1, both the squarate and oxalate act as bridging ligands adopting m1,2-bis-monodentate and bis-chelating coordination modes, respectively, connecting the Ho(III) ions forming the 1D ladder-like CPs, which generates hydrophilic pores intercalated guest water molecules. In 2, the squarate acts as bridging ligand with two coordination modes, m1,2-bis-monodentate and m1,2,3-tris-monodentate, connecting the Ho(III) ions forming 2D bi-layered MOFs. Intermolecular hydrogen bonds among the squarate, oxalate ligands and coordinated, guest water molecules provide the main force on the structural extension from their 1D ladder-like CP or 2D layered MOF to 3D supramolecular architectures. The solid-state adsorption spectra of 1 and 2 both show reversible color-changing images under illumination from an incandescent source/daylight to a LED light with a cellphone. Both 1 and 2 have an enhanced adsorption in the region around 450-480 nm (red line in the UV spectra), indicating that the ligand may enhance the absorption of whole coordination polymers.”

Round 2
Reviewer 1 Report
The authors have significantly improved the manuscript which is now acceptable for publication